# Central transcriptional regulator controls photosynthetic growth and carbon storage in response to high light

Seth Steichen[1], Arnav Deshpande [1], Megan Mosey[1], Jessica Loob[1], Damien Douchi[1], Eric P. Knoshaug[1], Stuart Brown[2], Robert Nielsen[2], Joseph Weissman[2], L. Ruby Carrillo [2] & Lieve M. L. Laurens [1] ✉

Carbon capture and biochemical storage are some of the primary drivers of photosynthetic yield and productivity. To elucidate the mechanisms governing carbon allocation, we designed a photosynthetic light response test system for genetic and metabolic carbon assimilation tracking, using microalgae as simplified plant models. The systems biology mapping of high light-responsive photophysiology and carbon utilization dynamics between two variants of the same *Picochlorum celeri* species, TG1 and TG2 elucidated metabolic bottlenecks and transport rates of intermediates using instationary $^{13}C$-fluxomics. Simultaneous global gene expression dynamics showed 73% of the annotated genes responding within one hour, elucidating a singular, diel-responsive transcription factor, closely related to the CCA1/LHY clock genes in plants, with significantly altered expression in TG2. Transgenic *P. celeri* TG1 cells expressing the TG2 CCA1/LHY gene, showed 15% increase in growth rates and 25% increase in storage carbohydrate content, supporting a coordinating regulatory function for a single transcription factor.

Carbon fixation in photosynthetic organisms follows a careful orchestration of photophysiological and metabolic phenomena, many of which are conserved across plants and crops, and are implicated in the global quest to increase agricultural yields[1,2]. Plants and algae adjust their physiology in response to saturating light conditions to optimize fitness in part by partitioning fixed carbon towards either storage molecules, such as starch, or anaplerotic metabolites facilitating increased growth rates. A central question arises on whether it is possible to re-direct carbon to storage compounds such as starch while avoiding an often-negative impact on growth rates. Three major physiological control systems have been implicated in crop yield improvements are plant growth, size, and/or seed yield through the targeted engineering of expression and function of key enzymes in these pathways[2]. Fewer reports document the untargeted, species-agnostic identification of transcriptional and physiological governing systems to counter the unexpected plant responses that have been

observed[3]. One such example is the discovery of a global regulatory transcription factor that was shown to modulate photosynthesis, seed yield, and nitrogen utilization efficiency, thus simultaneously impacting source and sink relationship in plants, often closely following diel cycling of light, temperature, and other environmental stimuli[3].

As simplified single-cell models of photosynthetic carbon allocation dynamics, biomass production by microalgae depends on robust cellular growth rates and compositional makeup upon harvest. Both aspects and their inter-relationships can be partially understood as functions of carbon flux rates and allocations downstream of carbon fixation through RuBisCO, into the Calvin Benson Bassham (CBB) cycle, and through central carbon metabolism towards final structural and storage molecules in photoautotrophic cells. Global bioprospecting projects have identified algae with unprecedented high growth rates under ideal conditions such as *Picochlorum spp.*[4–6] and *Chlorella ohadii*[7]. Comparing experimental metabolite tracking among

[1]Bioenergy Science and Technology Directorate, National Renewable Energy Laboratory, 15013 Denver West Parkway, Golden, CO 80401, USA. [2]ExxonMobil Technology and Engineering Co. (EMTEC), CLD286 Annandale, 1545 Route 22 East, Annandale, NJ 08801, USA. ✉e-mail: lieve.laurens@nrel.gov

algae (in particular *C. ohadii*), model flowering plants, i.e., *A. thaliana* and crops, i.e., *Z. mays*, showed that algae harbor increased flux through CBB metabolites, phosphoenolpyruvate (PEP), positioning the CBB cycle in a central node for increasing yields in both plants and algae[8]. The chlorophyte microalgae species, *P. celeri*, exhibits exceptional tolerance to high light and temperature by dominating mixed planktonic cultures in a high light (1500 μmol m$^{-2}$ s$^{-1}$) isolation strategy[6]. The strain designated as TG2 has one of the fastest reported specific growth rates for a microalgae species of 0.34 h$^{-1}$ (roughly translating to a 2 h doubling time) after rapid acclimation to high irradiance (1000 μmol m$^{-2}$ s$^{-1}$)[4,9]. High light selection also produced a closely related *P. celeri* isolate, TG1, that presents a significantly lower specific growth rate than TG2 of 0.22 h$^{-1}$. In combination, these two strains represent a powerful study system to elucidate genetic and metabolic diel carbon allocation control mechanisms.

Photoautotrophs, such as the model *A. thaliana*, integrate diel signals through transcriptional control over intrinsic circadian rhythms and in response to light levels[10–12]. The dehydration-responsive element (DRE)−binding protein 1/C-repeat binding factor (DREB1/CBF) family of transcription factors have been shown to respond directly to transcriptional changes in circadian regulators in plants[13], and overexpression of rice OsDREB1C confers production advantages across several yield phenotypes[3].

The identification of metabolic control points over carbon allocation from the photosynthetic source across the CBB and central carbon metabolism is important to define the fate of fixed carbon[14]. Accurate estimation of intracellular fluxes in the metabolic network (e.g., through Instationary metabolic flux analysis (INST-MFA[15], or fluxomics with $^{13}$C stable isotope tracking of metabolic carbon) is a critical component of carbon allocation optimization by identifying metabolic bottlenecks in cyanobacteria[15–21], as well as plants[22,23] and algae[8,24,25]. Increased bifunctional fructose-1,6/sedoheptulose-1,7-bisphosphatase (FBP/SBPase) and transketolase flux was shown to simultaneously decrease the oxidative pentose phosphate flux under light limiting conditions in *Synechocystis sp.* PCC 6803 overexpressing FBP/SBPase and transketolase[21]. In *Arabidopsis*, $^{13}$C fluxomics showed that increasing source strength by increasing light availability doubled carboxylation rates but also resulted in increased photorespiration[22]. The centrality of transcriptional and metabolic regulation offers opportunities for their engineering towards desirable traits including yields. To our knowledge, the phenotypic complexity of an organism's light and carbon assimilation response necessitates careful experimental deconvolution, for which a test system has not been reported before.

The objective of this work was to elucidate the genetic and metabolic regulating mechanisms governing growth, light response, and composition differences of two isolates, or variants, of *P. celeri*, TG1 and TG2, whose genomes are very closely related, but exhibit a consistently different physiological, and transcriptional phenotype. High-light (HL) stress, under uniquely controlled environmental conditions, was used to elicit the fast growth and high starch accumulation phenotype, in TG2 and TG1, respectively. Closely coordinated transcriptomics with fluxomics analysis allowed for deconvoluting an extensive covariance matrix of the physiological response to identify a global regulatory mechanism. We then demonstrated the impact of the identified regulator on the global genetic and photosynthetic carbon

allocation phenotype of an engineered strain that may have implications towards a translatable central control system in higher plants.

## Results and discussion

### Transcriptomic interaction analysis identifies a circadian transcription factor during high light response

The TG1 and TG2 isolated variants are designated the same species *P. celeri* on the basis of their exact match between 18 S rDNA sequences[6]. Genetic variation underlying phenotypic divergence between the strains was characterized by comparing whole genome shotgun sequence data from TG1 DNA against one copy of the phased diploid genome assembly, designated "p0" and available at NCBI accession codes JAACMV010000001.1 through JAACMV010000015.1 [https://www.ncbi.nlm.nih.gov/nuccore/1811192891]. Variant analyses indicated greater divergence from the reference TG2 assembly than originally suspected. Combined single nucleotide polymorphisms and insertions/deletions indicated nearly 2% divergence from the TG2 reference across the 13.7 Gb haploid genome length (Table 1). This sequence homology permits the shared species identity, but as clearly differentiated strains or cultivars. The number of variants (over 200,000 unique) with putative effects on gene encoding and regulatory regions precluded the possibility of assigning a genomic variant basis to the observed phenotypes (Supplementary Table 1).

To further deconvolute the genetic basis of the isolates' respective response to HL and phenotypic differences, we carried out gene expression analysis upon transition and acclimation to HL intensity. Both isolates were cultured under non-light-limited, dilute cell conditions (<0.3 μg/mL chlorophyll) allowing for homogeneous exposure of 1000 μE light (referred to as HL as described in materials and methods) without self-shading effects. We characterized temporal gene expression changes as cultures acclimated from low light (LL) to HL over the course of 4 h. The relative transcript abundances of TG1-WT and TG2-WT biological replicates differed significantly during this acclimation to HL growth conditions relative to the LL acclimated state established by 48 h of growth at 60 μE (Fig. 1). Both isolates demonstrated strong transcriptional responses to increased light with as much as 73% of annotated coding sequence transcript levels significantly changed after 1 h in HL ($p < 0.05$). The proportion of light responsive genes in *P. celeri* was larger than similarly treated photoautotrophs *Chlamydomonas reinhardtii*, *Chromochloris zofingiensis*, and *Arabidopsis thaliana* (Supplementary Table 2)[26–29]. This drastic shift could be attributed to the extremely small and streamlined *P. celeri* genome being largely devoted to light response. Large magnitude changes in transcript levels were limited to a small set representing <9% of transcripts (L2FC > 3).

To specifically look for unique and differential gene expression behavior in the light response of both isolates, several different analyses were carried out to isolate the interaction effect between light and strain for the 1 h time point. The majority of significantly changed transcript levels after 1 h in HL (|L2FC| > 1.0; $p < 0.05$) were down regulated relative to the LL levels in each of the wild type strains (Fig. 1A, B). The most highly induced transcript at this time was the same for both strains, encoding a carotene biosynthesis-related protein (CBR; EMRE3EUKT4063781). CBR expression likely reflects a strong response of the photosynthetic machinery to dissipate excess light until the physiological HL acclimation is complete, as was similarly implicated in another high light tolerant green algae, *C. ohadii*[30]. This theme of transcripts being similarly regulated between the strains was carried throughout the transcriptome. In fact, there was not a single transcript which was changed in the opposite direction between TG1 and TG2 after applying the same fold change and significance thresholds (Fig. 1C). With a carefully-designed likelihood ratio test on the full transcriptome, we were able to identify several transcripts that exhibited a statistically significant interaction effect in their expression response to high light transition and genetic background

**Table 1 | *Picocholorum celeri* TG1 genomic variants summary**

| Treatment | SNPs | INDELs | total | % genome |
|---|---|---|---|---|
| Unfiltered | 351,259 | 44,390 | 395,649 | 2.89 |
| Lenient | 328,218 | 38,541 | 366,759 | 2.68 |
| Strict | 236,538 | 23,812 | 260,350 | 1.90 |
| TG2 diploid subtraction | 194,648 | 23,567 | 218,215 | 1.59 |

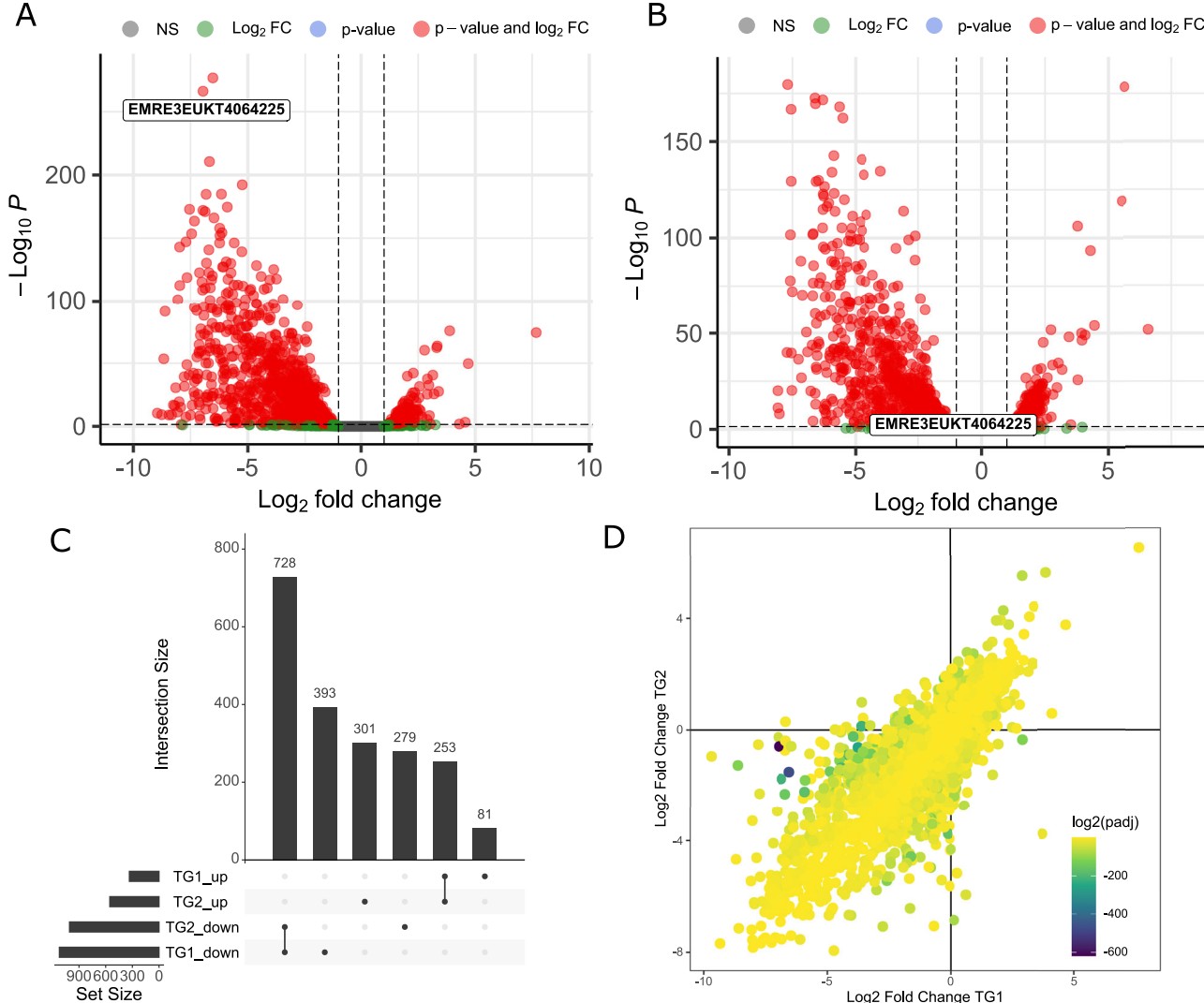

**Fig. 1 | Gene expression variation between TG1-WT and TG2-WT upon exposure to high light intensity.** RNA-seq data collected from wild type *P. celeri* cultures after 1 h of exposure to HL conditions following acclimation to LL (T = 0). Changes in transcript abundances of (**A**) TG1 and (**B**) TG2 cultures expressed as Log2 of the fold changes between 1 h high light and steady state low light growth. Dashed lines represent threshold values for fold change and *p*-values (L2FC(1) and 0.05, respectively) used to identify significant gene sets, as determined by a two-sided Wald test using Benjamini–Hochberg multiple testing adjustment. The putative CCA1 transcript is labeled in each with its transcript ID number, and not significant (NS) genes colored in gray. **C** Comparison of significant differentially expressed gene sets between TG1 and TG2. The sets identified in above panels were separated into up or down regulated groups and all combinations of comparison are represented in the lower matrix with the number of members shown in the above bars. **D** Statistical analysis to deconvolute *P. celeri* TG1 and TG2 isolates' transcriptome. The results of a two-sided likelihood ratio test determining the significance of each gene to a model isolating the interaction term between high light exposure time and strain is displayed as color scaled *p*-values over the fold change observed for each gene. All data shown are derived from independent biological replicate cultures (*n* = 3). Source data are provided as a Source Data file.

(Fig. 1D). The single most significant gene identified in this analysis (EMRE3EUKT4064225), was annotated as a transcription factor encoding a homolog of late elongated hypocotyl/circadian clock associated like (LHY/CCA) and was strongly separated from the rest of the transcriptome. The translated protein sequence indicated a close relationship with *A. thaliana* homologs for CCA1 and LHY containing the conserved domain related to the SANT family of MYB-like DNA binding sequences (Supplementary Fig. 1) hereafter referred to as *Picochlorum* CCA1. The variation in TG2 and TG1 CCA1 gene transcription may be in part due to sequence variant sites apparent in an alignment of the genomic loci (Supplementary Fig. 2A), including a 9 bp insertion in the TG2 promoter region 590 bp upstream of the translation start site as well as a deletion and insertion event within 170 bp downstream of the stop codon, which may lie beyond the 3' UTR as RNA-seq read data coverage extends to -165 bp. The coding sequence of the TG1 CCA1 gene contains up to 20 predicted non-synonymous sequence variants with respect to the TG2 genomic locus. Only two changes in amino acid residues lie within the single identifiable conserved functional domain region, neither of which alter the conserved residues defining the SANT domain structure (Supplementary Fig. 2B). We hypothesize that this transcription factor acts as one of the core regulatory factors that underly the respective observed strain phenotype (Fig. 1D).

**Regulatory genome-wide transcriptional analysis suggests control over carbon metabolism and circadian response**
We tested the impact of the genome-wide regulatory CCA1 function in *P. celeri* as related to both growth and carbon allocation dynamics by creating a genetically engineered variant of TG1 with the TG2-WT CCA1 gene elements transformed into a TG1-WT background using an

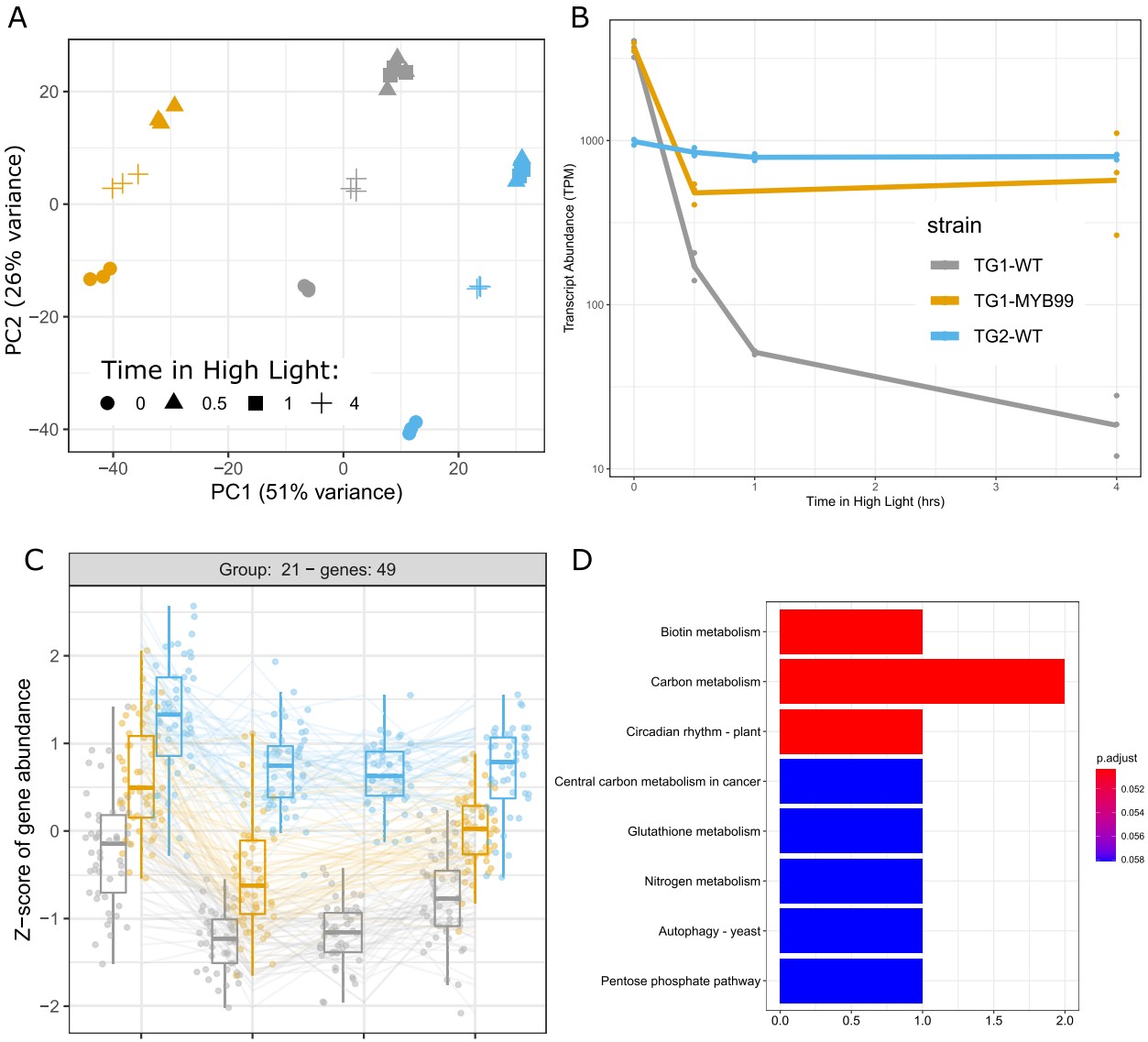

**Fig. 2 | Transcriptional responses of *Picochlorum celeri* TG1 and TG2 isolates and engineered TG1-MYB99 strain, in response to high intensity light shift.** **A** The relative position of each sample is displayed based on a principal component analysis scores plot of the regularized log transform data of the 500 most highly expressed transcripts. **B** CCA1 transcriptional dynamics over the acclimation time course. Relative transcript levels were computed following alignment to a reference with a single CCA1, and therefore represent the total pool of TG1 and TG2 transcript isoforms. **C** Time course transcript dynamics of genes clustered based on their similarity to CCA1 expression are displayed by their deviation from median transcript levels of the cluster across the time course with boxplots representing distributions of expression level variances for genes averaged from independent biological replicate cultures (*n* = 3) centered on the median with lower and upper bounds on the 25th and 75th percentiles, respectively. Lower and upper whiskers indicate minimum and maximum values up to 1.5 times interquartile range in each direction. The corresponding bar plot (**D**) of enriched functional annotation categories indicates the number of genes with each annotation type and colored by the significance of their over-representation compared to the remaining background genome as computed using a hypergeometric distribution with Benjamini–Hochberg multiple testing adjustment.

expression vector containing the TG2 CCA1 coding sequence under control of the TG2 native promoter and terminator regions (Supplementary Fig. 3). The cell line with the highest specific growth rate in HL conditions was selected from a screen of transformants for further analysis and designated TG1-MYB99. After genomic integration and both RT-PCR and phenotypic confirmation of the transformant line, we included this strain alongside the WT isolates for further regulatory and response analysis of the mutant. The insertion site of the TG2-like CCA1 transgene in the TG1-MYB99 line was associated with NIPA2 magnesium transporter coding sequences (Supplementary Table 3). It was not possible to map unambiguously due to multiple NIPA2 gene copies, however, evaluation of transcript levels indicated that the insertion likely disrupted the EMRE3EUKT4068533 gene which was

compensated for by elevated transcription rate of the remaining copies in the TG1-MYB99 genome (Supplementary Fig. 4).

A four-hour time course transcriptomics analysis during transition from LL to HL was conducted to determine the transcriptional regulatory alterations in the TG1-MYB99 strain. All strains demonstrated a biphasic acclimation to the increased light intensity, with the largest shift in gene expression occurring 30 min and 1 h after HL and gene expression trending back towards LL levels by the fourth hour (Fig. 2A). The differential transcription dynamics of CCA1 were further validated by light-shift time course results, with a persistent down-regulation in the TG1-WT cultures while TG2-WT maintained the intermediate level observed under LL illumination (Fig. 2B). The CCA1 gene reaches maximum transcriptional expression around dawn in

both land plants and the marine microalgae, *Ostreococcus tauri*, when entrained to diel light cycles[10]. We postulate that TG2-WT strain may be a circadian mutant relative to the TG1-WT strain, as the rapid suppression of CCA1 transcription in response to HL resembles patterns reported at dawn in other systems. The time course of CCA1 gene expression dynamics of the TG1-MYB99 strain confirm the successful transfer and expression of TG2 gene elements, starting with high-expression levels as for WT TG1 and upon HL transition shifting to follow the continued high expression as observed for TG2 (Fig. 2B). The global transcriptomics response over the 4 h time course similarly positions the TG1-MYB99 strain as intermediate between TG1 and TG2, with a significant fraction of the transcriptome following closely the expression patterns observed in TG2 (Fig. 2C). Over-representation statistical analysis of the set of 49 genes that closely matched CCA1 expression dynamics between the cell lines implicated important functions related to biotin metabolism (fatty acid biosynthesis, amino acid catabolism), broad central carbon metabolism, and circadian rhythms (Fig. 2D). Further representation of transcriptional dynamics of the respective major pathways' gene expression is included as heat maps (Supplementary Figs. 5 and 6).

Only a small set of genes shared transcriptional dynamics with CCA1 among the strains in response to shifting from LL to HL (Fig. 2C). LHY, CCA1, and related transcription factor binding sites (TFBS) were significantly enriched in the upstream regulatory regions of the 49 cluster 21 genes relative to the regulatory sequences of all other genes across the *P. celeri* genome (Supplementary Data 1). Similarly, TFBSs for REVEILLE family transcription factors were enriched at nearly identical levels. This is likely to represent consolidated functionality of the CCA1 gene in *P. celeri*, which has a severely reduced quantity of circadian clock-related genes relative to plants like *A. thaliana*[11]. Expression of genes annotated with functions related to carbon metabolism was enriched among the set containing CCA1 (Fig. 2D), and widespread alteration in expression of genes related to carbon fixation was observed in the mutant line (Supplementary Fig. 6).

### Instationary metabolic flux analysis confirms metabolic control nodes for central carbon allocation to carbohydrates

To assess the respective biomass composition response to light transition, we quantified the primary macromolecular biochemical composition of harvested algal biomass after acclimation to low or high light. Experimental biomass composition data was measured for protein, carbohydrates, and FAME. However, experimentally measured composition does not often satisfy mass balance[31] and must be corrected due to factors such as high ash content and underestimation of components. To account for the missing fractions of biomass that we did not measure experimentally, chlorophyll content was assumed from *P. celeri*[6], while DNA and RNA content was assumed from previous work in algae[24]. FAME and carbohydrate content was corrected and subsequently all data were scaled to satisfy mass balance as described in Supplementary Information, INST-MFA assumptions. Carbon allocation varied dramatically between the cell lines in response to HL conditions, with TG1-WT accumulating as much as 44.9% of its total biomass as carbohydrate, while TG2-WT only reached 16.0% after 48 h of growth in HL. The TG1-MYB99 mutant line cells, remarkably, apportioned over 56% of their biomass to carbohydrate content after the same 48-h acclimation period to HL growth (Table 2). This phenotype contrasts with the intermediate expression level of CCA1 in the mutant line with respect to the WT *P. celeri* cultures. To quantify metabolite fluxes through central nodes of carbon assimilation and storage, we applied $^{13}C$ labeling enabled INST-MFA of TG1 and TG2, and the TG1-MYB99 mutant. This quantitative analysis aims to elucidate key differences in metabolism. Because $^{13}C$-labeled $CO_2$ is challenging to introduce in an immediate and step-change manner (instantaneous change to $^{13}C$ from $^{12}C$ for subsequent carbon fixation) that allows for following the $^{13}C$ tracer over a time course[15], we used a bolus of $^{13}C$

labeled bicarbonate (1 g $L^{-1}$ $NaH^{13}CO_3$) in a series of physiologically-controlled experiments (see Supplementary Information for more in depth discussion and experimental results). A step-change is crucial since several fast labeling metabolites such as those involved in the CBB cycle incorporate the label in the order of a few seconds[8,15,16]. The sampling time points were chosen from 30 s to 10 min such that labeling dynamics for fast labeling metabolites were able to be fully captured. To prevent metabolite leakage by quenching methods such as cold methanol quenching[32], we rapidly filtered cells and quenched in a liquid $N_2$ bath within 10 s of the timepoints.

The INCA 2.0 software package[33] was used to calculate the flux values based on time course labeling data (Supplementary Data 2), biomass composition (Supplementary Table 4), and specific growth rate (Supplementary Table 5). The metabolic network and atom transitions were taken and adapted from previous work in algae[8,24,34] and is reported in Supplementary Data 3. The same metabolic network was used for all three strains and includes compartmentalization and dilution parameters to account for inactive metabolite pools or metabolite channeling[17]. The glyoxylate shunt is not modeled since it is unlikely to exist based on the *P. celeri* genome. BLAST results revealed that the genome for *P. celeri* has poor matches for the genes malate synthase and isocitrate lyase that are known to facilitate the glyoxylate shunt in algae. There is evidence that the glyoxylate shunt has been lost in several clades of algae[35]. The fitted models estimated $CO_2$ uptake from the sink demands and was not supplied as an input parameter. TG1, TG1-MYB99, and TG2 had a $CO_2$ uptake of 5.96, 6.99, and 9.59 mmol gBiomass$^{-1}$ h$^{-1}$ respectively. The model fits were assessed by evaluating the SSR. The best fitted models for TG1, TG1-MYB99, and TG2 had SSR = 226 [160, 237.8], SSR = 244.2 [213.6, 302.2], and 297.3 [213.6, 302.2]. To identify differences in flux partitioning at different metabolic nodes, the fluxes were normalized to $CO_2$ uptake rate as shown in Fig. 3. (Fluxes and their 95% confidence intervals are available in Supplementary Data 4; measured and fitted mass isotopomer distribution (MID) profiles are plotted in Supplementary Figs. 7–9.)

Although there are variances between the net fluxes in TG1, TG1-MYB99, and TG2, key differences in metabolic points of control can be identified in Fig. 3, which shows the flux after normalizing to uptake of 100 units of $CO_2$. TG1 and TG1-MYB99 have a high carbohydrate content which is consistent with the higher flux from F6P in the chloroplast to G6P followed by conversion to G1P and subsequently carbohydrates compared to TG2. The normalized flux from 3PGA as the first product formed after carbon fixation to F6P is not significantly different in the three strains, and a flux difference is only observed in subsequent conversion of F6P to carbohydrates. The conversion of F6P to G6P is catalyzed by the reversible G6P isomerase (EMRE3EUKT4065295) followed by transfer of the phosphate group to position 1 to form G1P catalyzed by phosphoglucomutase (EMRE3EUKT4067541). This is subsequently converted to ADPG by G1P-adenylyl-transferase (EMRE3EUKT4064735, EMRE3EUKT4063059). Transcriptomics suggests that the expression of these enzymes is at least 2-fold higher in TG1 and TG1-MYB99 compared to TG2. This higher expression contributes to the higher flux to carbohydrates in these strains and may suggest that their expression is co-regulated. Expression of PGI and G1P adenylyl-transferase is similar in both TG1 and TG1-MYB99 whereas, in the case of PGM, TG1 has ~2.5-fold higher expression while TG1-MYB99 has a ~4 fold higher expression compared to TG2. This indicates that CCA1 expression further increases carbohydrate accumulation through the control over PGM. This is in line with the observations in *C. reinhardtii* where increased PGM1 expression was implicated in starch accumulation[36]. Increased flux through over-expression of these enzymes diverts flux carbon away from regeneration of precursors for photosynthetic carbon fixation via the CBB cycle and may result in TG1 and TG1-MYB99 having a lower net carbon fixation compared to TG2.

**Table 2 | *Picocholorum celeri* strains growth rate and biomass composition for the same experiments**

| Strain | Light | Growth Rate (hr⁻¹) | FAME (%) | Protein (%)ᵃ | Carbohydrate (%) | DNA (%) | RNA (%) | Chl A (%) |
|---|---|---|---|---|---|---|---|---|
| TG1 | LL | 0.110 ± 0.038 | 18.8 ± 0.2 | 55.9 | 10.2 ± 0.2 | 3.9 | 8.7 | 2.6 |
| TG2 | LL | 0.079 ± 0.039 | 21.6 ±1.3 | 49.0 | 8.6 ± 0.8 | 4.8 | 10.7 | 3.2 |
| TG1-MYB99 | LL | 0.048 ± 0.037ᵇ | 19.6 ± 0.1 | 55.0 ± 0. 3 | 10.4 ± 0.4 | 3.8 | 8.6 | 2.6 |
| TG1 | HL | 0.220 ± 0.079 | 11.8 ± 0.2 | 29.9 ± 0.1 | 44.9 ± 0.3 | 3.4 | 7.66 | 2.3 |
| TG2 | HL | 0.323 ± 0.043 | 16.9 ± 0.3 | 50.8 | 16 ± 1.1 | 3.9 | 8.6 | 2.6 |
| TG1-MYB99 | HL | 0.254 ± 0.041 | 9.5 ± 0.3 | 21.0 ± 3.0 | 56.1 ± 2.7 | 3.4 | 7.6 | 2.3 |

Values are expressed as a percentage of total biomass, reported on an ash free basis, growth rates were calculated based on total organic carbon increase over time. All data are shown as mean ± standard deviation for at least triplicate biological samples collected under consistent physiological conditions.
ᵃMissing error values indicate single replications of protein analysis due to limiting biomass.
ᵇGrowth rates of TG1-MYB99 in LL higher calculated on chlorophyll basis (0.064 ± 0.003).

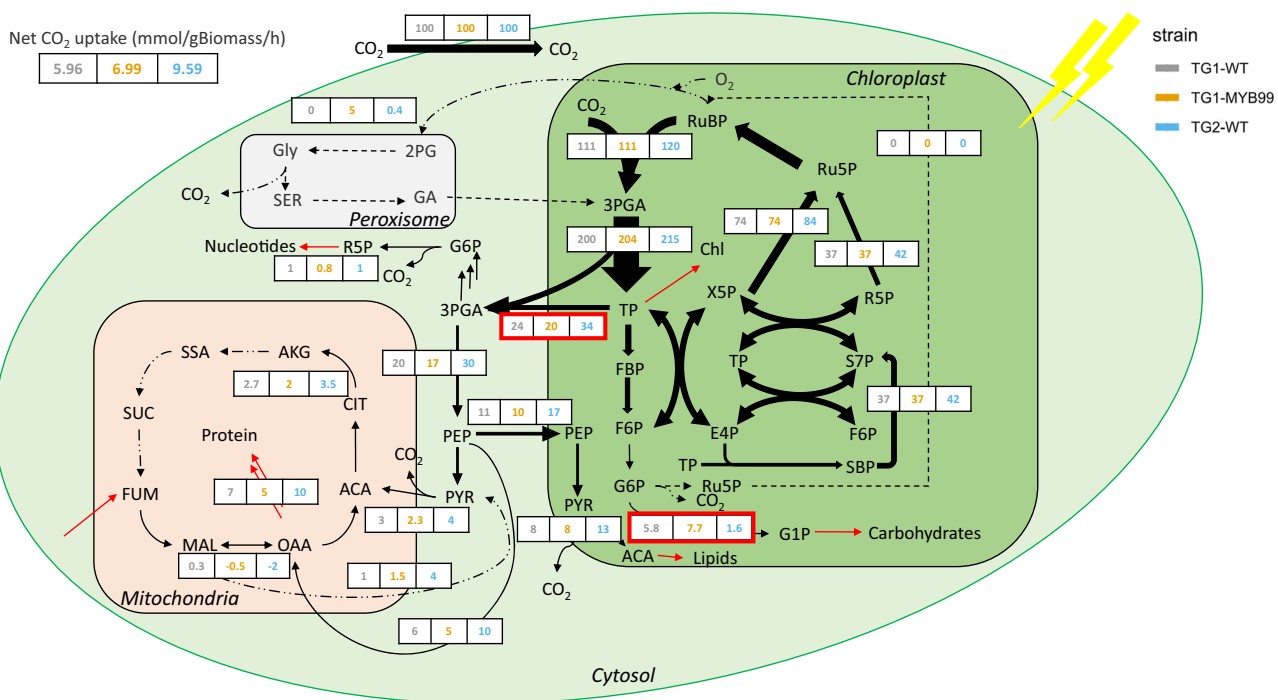

**Fig. 3 | Flux maps determined in TG1, TG1-MYB99, and TG2 under continuous HL conditions (1000 μE).** For each reaction shown, the net fluxes are normalized to 100 units of $CO_2$ taken up. All flux values and confidence intervals are given in Supplementary Data 4. Estimated net $CO_2$ uptake rates for TG1, TG1-MYB99, and TG2 are 5.96, 6.99, and 9.59 mmol gBiomass⁻¹ h⁻¹ respectively (derived from flux models, using $n = 3$ biological replicated experiments). Ribulose 1,5-bisphosphate (RuBP), 3-phosphoglycerate (3PGA), Triose phosphate (TP), Fructose 1,6-bisphosphate (FBP), Fructose 6-phosphate (F6P), Glucose 6-phosphate (G6P), Glucose 1-phosphate (G1P), Ribulose 5-phosphate (Ru5P), Erythrose 4-phosphate (E4P), Ribose 5-phosphate (R5P), Xylulose 5-phosphate (X5P), Sedoheptulose-7- phosphate (S7P), Sedoheptulose-1,7-bisphosphate (SBP), Phosphoenolpyruvate (PEP), Acetyl-CoA (ACA), Pyruvate (PYR), Glycine (Gly), Serine (SER), 2-phosphoglycolate (2PG), Glycolic acid (GA), Citrate (CIT), Alpha-ketoglutarate (AKG), Succinic semialdehyde (SSA), Succinate (SUC), Fumarate (FUM), Malate (MAL), Oxaloacetate (OAA). Black arrows represent intracellular flux while red arrows represent fluxes to/from biomass sink. The width of the black arrows is indicative of the magnitude of the flux. The red boxes highlight reactions catalyzed by phosphoglucomutase (PGM) and triose phosphate transporter (TPT3) which are implicated in the differing biomass composition phenotypes.

The different biomass composition of the strains (Table 2 and Supplementary Table 6) necessitates a distinct carbon allocation. TG2, which has a higher protein content demands more precursors from the TCA cycle whereas TG1 and TG1-MYB99 produce carbohydrates and require higher glycolytic flux in the chloroplast. Further, lipid production occurs in the chloroplast from PEP imported from the cytosol. The precursor demands are reported in Supplementary Table 7. In green algae, the upper half of glycolysis is chloroplastic while the lower half is in the cytosolic compartment[37]. Thus, metabolite transport across the chloroplast where carbon is fixed has a crucial link to biomass composition. We hypothesized that TG2, which has a higher protein and lipid content will have a higher flux to the cytoplasm through the triose phosphate transporter (EMRE3EUKT4065742) to supply metabolic precursors for amino acid biosynthesis and back into the chloroplast through the phosphoenolpyruvate transporter (EMRE3EUKT4068085) to supply acetyl-CoA for lipid biosynthesis and PEP for synthesis of aromatic amino acids by the shikimate pathway. This is confirmed by the flux data which shows that TG2 has a 40% and 70% increased transport to the cytoplasm compared to TG1 and TG1-MYB99, respectively, whereas the transport back into the chloroplast is 60% higher in TG2 compared to both TG1 and TG1-MYB99. This is consistent with recent work in *C. reinhardtii* where TPT3 null mutants showed a reduced growth phenotype coupled with an increase in carbohydrate accumulation[38]. Interestingly, transcriptomics revealed

that expression of the triose phosphate transporter (EMRE3EUKT4065742) had exactly the opposite trend with the lowest abundance in TG2 and highest in TG1-MYB99, possibly pointing to a post-transcriptional regulation. In the case of the PEP transporter (EMRE3EUKT4068085), TG1-MYB99 exhibited significantly increased transcript abundance from the similar levels observed in TG2 and TG1. This could indicate that in *P. celeri*, transport between the chloroplast and cytoplasm is not limited by transcript abundance and may be influenced at the protein level or through post-translational modifications. These observations are consistent with carbohydrate accumulation in the form of starch, as a metabolic carbon storage and overflow mechanism, and is a key feature of diel and circadian cycles of plants and algae such as *A. thaliana* and *C. reinhardtii*[39,40]. Non-additive alterations in CCA1 and LHY expression have been linked to improved agronomic traits in allotetraploid *A. thaliana* hybrid plants that presented increased starch, chlorophyll content, and growth compared to parental lines[41].

In addition to these nodes, carbohydrate accumulation as starch, as expected, was shown to have a significant effect on $^{13}C$ labeling dynamics of amino acids. A reduction in amino acid pool sizes was observed for most amino acids in TG1 and TG1-MYB99 (Supplementary Fig. 10). Even though it takes a higher flux to achieve the same labeling for a larger pool size, all the amino acids showed faster incorporation of $^{13}C$ label in 10 min in TG2 (Supplementary Fig. 11). This higher flux to amino acids contributes to the protein rich biomass composition of TG2 and could be necessary for faster growth.

## Tricarboxylic acid (TCA) metabolism regulation in *P. celeri* suggests incomplete cycle upon high light acclimation

The fluxes through the TCA cycle in all the strains show no significant flux from alpha-ketoglutarate (AKG) to succinate and subsequently to fumarate (Fig. 3, Supplementary Data 4). This is also reflected in the labeling of succinate (Supplementary Figs. 7–9) which does not label significantly in the 10 min timescale. This observation indicates that *P. celeri* may prefer to operate a linear branch from fumarate to AKG under HL acclimated conditions. This has been previously observed in fast growing cyanobacteria such as UTEX 2973 and *S. elongatus* 7942[16,25,42] and may be due to very low expression of Succinyl-CoA ligase which catalyzes the synthesis of succinate from Succinyl CoA (EMRE3EUKT4065470). The $^{13}C$-labeling dynamics of other TCA cycle metabolites measured such as fumarate, malate, citrate, and AKG indicate significant labeling in all strains compared to succinate further indicating that a complete TCA cycle may not be essential under HL conditions in *P. celeri* (Supplementary Figs. 7–9). This could potentially be a mechanism in fast growing strains under high light conditions to prevent loss of $CO_2$ from the conversion of AKG to succinate via the succinyl-CoA as energy demands in the form of NADH or ATP are met by light-dependent reactions.

Even though these three strains seemingly operate an incomplete TCA cycle under HL acclimated conditions, there exist a few key differences between protein-rich TG2 and carbohydrate-rich TG1 and TG1-MYB99. Since several amino acids draw flux from the TCA cycle, TG2 drives roughly a two-fold normalized flux compared to TG1 and TG1-MYB99 (Fig. 3, Supplementary Data 4). The higher flux to the TCA cycle results in a higher loss of $CO_2$ in TG2 due to decarboxylation. Interestingly, TG2 can increase RuBisCO carboxylase and PEP carboxylase flux to recapture the $CO_2$ lost as evidenced by the normalized flux through RuBiscCO and PEP carboxylase (Fig. 3). Higher TCA cycle flux is achieved by a higher transcription of the mitochondrial dihydrolipoyllysine-residue acetyltransferase component of pyruvate dehydrogenase complex (EMRE3EUKT4066919, identified by sequence similarity by BLAST) in TG2 compared to TG1 and TG1-MYB99. Expression of EMRE3EUKT4066919 is ~50% and ~86% higher in TG2 compared to TG1 and TG1-MYB99 respectively. In addition to a higher flux to the TCA cycle via pyruvate dehydrogenase, TG2 shows a significantly higher anaplerotic TCA cycle flux than TG1 and TG1-MYB99 via the PEP carboxylase (EMRE3EUKT4066235) reaction which converts PEP to oxaloacetate. Although TG2 has the highest anaplerotic flux to meet the TCA cycle's energy demands, PEP carboxylase expression is the highest in TG1-MYB99, followed by TG2, and then TG1. This may be due to regulation of PEP carboxylase at the post translation modification level as reported previously in algae and plants[43,44]. Malate and fumarate label rapidly in all strains indicating the activity of PEP carboxylase (Supplementary Figs. 7–9). Fumarate hydratase or fumarase (EMRE3EUKT4065827), which can reversibly convert fumarate to malate, is active in all the strains and characterized by a high exchange flux resulting in rapid labeling of fumarate (Supplementary Figs. 7–9) and has the highest expression in TG2, followed by TG1 and TG1-MYB99.

Interestingly, the higher flux into the TCA cycle from PEP carboxylase in TG2 results in a net flux from oxaloacetate to malate in TG2, which is then catalyzed back to pyruvate by NADP-dependent malic enzyme. This results in a cycle that causes the net production of NADPH by the malic enzyme reaction. Whereas, in case of TG1 and TG1-MYB99, malate dehydrogenase catalyzes the forward reaction from malate to oxaloacetate and malic enzyme flux is negligible. In terms of transcription of NADP-dependent malic enzyme (EMRE3EUKT4065957), expression is similar in TG1 and TG2, while it is 23% lower in TG1-MYB99. Metabolism of malate is influenced by the redox status of cell as well as the cellular requirements for ATP and NADPH of the cell[45–48]. These differences indicate that TG2 may have a higher NADPH demand which necessitates a higher malic enzyme flux especially since the flux through the oxidative pentose phosphate pathway, which also produces NADPH is negligible. It is not surprising that this is the case since several fast-growing cyanobacteria and algae have been found to have negligible flux through this pathway which results in loss of $CO_2$[16,18]. A comparison of the absolute and normalized fluxes for strains in this study and previous work is algae and plants is shown in Supplementary Figs. 12 and 13.

## Estimation of ATP and NADPH demand in *P. celeri* supports cellular energetics balance

Due to distinct biomass composition and metabolic demands for these strains, the cellular ATP and NADPH demands were calculated based on metabolic fluxes from our model (Supplementary Data 5). The ATP and NADPH demands for biosynthetic reactions were calculated based on reactions listed in Supplementary Table 7. The ATP maintenance cost was assumed to be 2.85 mmol $g^{-1}$ $h^{-1}$ as reported in *C. reinhardtii*[49] since data is unavailable for *P. celeri*. Our estimates show that TG2 consumes ATP/NADPH in the ratio 1.58 whereas this number is 1.71 and 1.69 for TG1 and TG1-MYB99 respectively. These estimates are close to the ATP/NADPH demand of 1.6 that has been suggested previously[50]. Our data may indicate that the malic enzyme shuttle from malate to pyruvate has a higher flux in TG2 (Fig. 3), resulting in the net production of NADPH, acts as an additional overflow pathway to supply NADPH to balance cellular energetics[51], and is essential in TG2 to meet its higher NADPH demand.

Furthermore, the ATP/NADPH ratio is of great importance since an imbalance between the synthesis and consumption of ATP and NADPH is directly linked to photosynthesis[52]. Linear electron flow (LEF) produces ATP/NADPH in the ratio 1.29 based on the coupled electron-proton reactions that power ATP synthase to generate 3 ATP per 14 protons pumped[53]. The CBB cycle, on the other hand, requires a higher demand for ATP/NADPH in the ratio 1.5[54]. This imbalance in production and consumption has been suggested as a point of engineering in photosynthesis and forms the basis of the argument that cyclic electron flow (CEF), or an alternative flux of electrons to generate more ATP, is important to balance the ATP/NADPH synthesis and consumption[54–56] with energy demands required to respond to environmental perturbations. Based on these estimates, we hypothesize

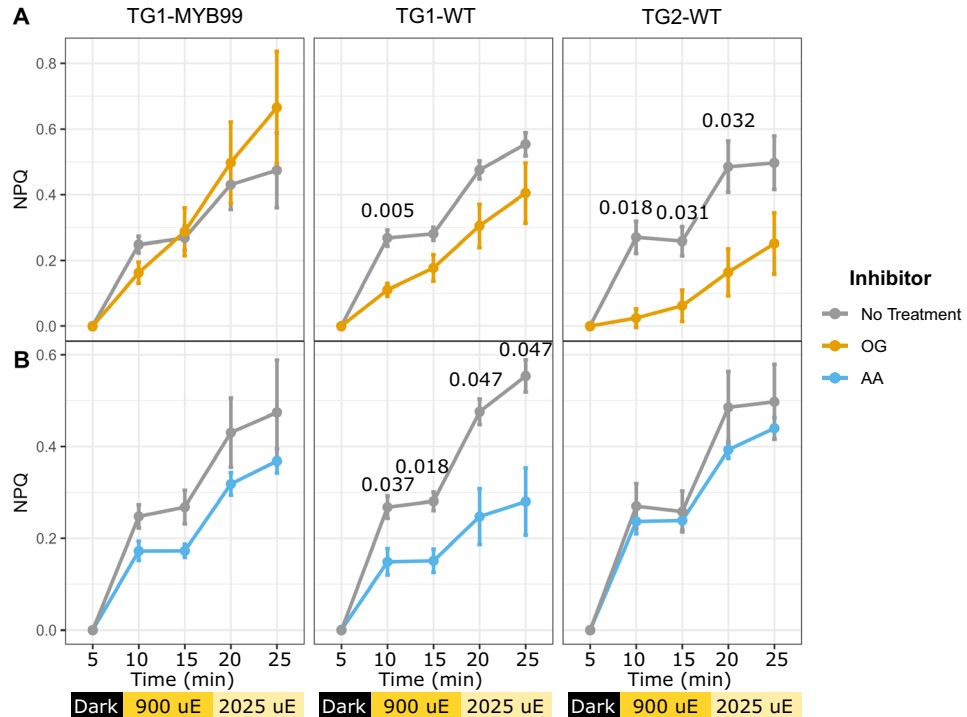

**Fig. 4 | Non-photochemical quenching (NPQ) responses of high light acclimated cell cultures.** Replicate cultures were sampled and treated with inhibitors reported to interfere with different NPQ mechanisms including (**A**) octyl gallate (OG) and (**B**) antimycin A (AA). Data points are presented as mean values ± SD of biological replicate cultures ($n = 3$). Asterisks indicate significant differences between inhibitor treated NPQ measures and untreated control values as determined by a two-sided $T$-test, with $p$-values displayed above each significantly different comparison ($p < 0.05$) and insignificant $p$-values omitted.

that TG1 and TG1-MYB99 have a higher cyclic/alternative electron flow compared to TG2. To test this, we utilized photophysiology experiments which enable the characterization of electron flow in the photosynthetic apparatus.

Acclimation responses of the photosynthetic apparatus and related mechanisms can often underly the ability of photoautotrophs to resist abiotic stresses including excessive light intensities. *P. celeri* TG1 and TG2 cells exhibited remarkably low non-photochemical quenching (NPQ) response after acclimation to high light, and particularly when measured using a pulse amplitude modulated (PAM) fluorometry instrument under 1000 µE actinic light. We developed custom protocols using a fast repetition rate fluorometer (FRRf) that could deliver much higher actinic light intensities to induce quenching responses (Supplementary Fig. 14). Both wild type strains and the TG1-MYB99 mutant exhibited similar NPQ responses to increasing actinic light, only reaching ~0.5 following complete saturation of photosystems with 2025 µE of background illumination (Fig. 4). Other algae species and plants like *A. thaliana* can reach >2.0 NPQ calculated on the classical scale[57] that does not have a specified upper limit, as opposed to the PhiNPQ framework that cannot exceed 1[58].

We probed the cells' photophysiology further by applying inhibitors of documented mechanisms for alternative electron flow. The plastid terminal oxidase (PTOX) protein acts in many photoautotrophs to oxidize plastoquinone (PQ) pools, and its activity can be inhibited by the application of octylgallate (OG)[59,60]. TG2-WT cultures demonstrated a reliance on PTOX activity based on significant suppression of NPQ response across most light levels when treated with OG, while TG1-WT exhibited a minimal impact from the inhibition (Fig. 4A). This may be indicative of a higher capacity of TG2 cells to respond rapidly to extreme light intensity increases, as PTOX can directly relieve PQ reduction pressure but at the cost of ATP and NADPH production along with reactive oxygen species generation[61].

The activity of PTOX may be unlikely to facilitate long term acclimation to extreme light, as its oxidation of over-reduced plastoquinone pools comes at the cost of harmful superoxide generation by a side reaction[62]. Alternatively, CEF does offer a sustainable mechanism to relieve over-reduction of the photosynthetic electron transport system by altering lumen proton gradients effecting pH sensitive mechanisms like NPQ, and proton motive force. We found that only TG1-WT cells significantly responded to antimycin A (AA) treatment during NPQ experiments (Fig. 4B), indicating an appreciable level of CEF activity. The AA sensitive CEF pathway has been observed across plant and algae species, with CEF deficient mutant in *A. thaliana* displaying impaired NPQ responses[63]. The *P. celeri* genome contains homologous sequences to two key proteins in the AA sensitive CEF mechanism. Transcription of a PGRL1B annotated gene (EMRE3EUKT4068276; Supplementary Fig. 15), a transmembrane protein thought to be present in thylakoids, was suppressed in TG2-WT during HL acclimation, while TG1-WT maintained high levels of expression. This protein was shown to moderate the electron transfer from ferredoxin back to the PQ pool in plant systems, and implicated in sustaining CEF[64]. Remarkably, nearly all the photophysiology metrics derived from FRRf measurements did not vary significantly among the cell lines. One exception was observed in the second order electron transfer time constant being significantly lower in the TG2 cells than TG1 (Supplementary Table 8; $p = 0.0354$) at $269.00 ± 8.66$ µs and $421.33 ± 59.47$ µs, respectively. This rate constant was calculated at an intermediate value of $382.00 ± 103.36$ µs in the TG1-MYB99 cells, though the difference in rate from each wild type cell line was not statistically significant.

The *P. celeri* wild type and mutant cell lines discussed herein represent a unique and accessible study system for interrogating and testing relationships between the critical agronomic yield traits of biomass productivity and carbon allocation and storage. The majority of algae strains with documented productivities high enough to

support industrial biofuel operations require induction of stress conditions to elicit accumulation of either lipid or carbohydrate products to reach maximum valorization[65]. This approach does reduce growth and productivity if applied too long, and so necessitates careful planning, monitoring, and timely control. Several studies have shown that genetic engineering and other approaches to control specific aspects of plant and algae physiology can improve yield traits while minimizing inhibition of growth or photosynthetic efficiency[66,67]. TG1-WT and TG2-WT cells provide evidence that growth rates and carbon allocation dynamics can be influenced strongly by a central regulatory factor in the form of CCA1 in the context of acclimation responses to HL exposure. The mixed and opposing phenotypes observed in the TG1-MYB99 mutant cell line are indicative of complex and indirect regulation. While these can be partially interpreted by gene clustering and transcription factor binding site (TFBS) analyses, more powerful and direct tools were needed for evaluation of biochemical outcomes of the genetic manipulation. Fluxomic results, along with requisite assessments of biomass composition, provide a powerful method for evaluating the effects of genetic engineering on some of the most important phenotypic factors affecting *P. celeri* productivity and valorization potential.

## Methods

### Growth conditions

Two wild type *Picochlorum celeri* strains, TG1 and TG2, were originally isolated from environmental samples by selecting for fast growth in high light intensity as previously described[6]. Cell line stocks were maintained in cryogenic storage, and revived as needed in a modified artificial seawater medium (MASM), containing the following: 10 mM $MgSO_4$, 10 mM $NaNO_3$, 10 mM Tris-HCl, 8 mM KCl, 2 mM $CaCl_2$, 0.5 mM $NH_4Cl$, 0.37 mM $KH_2PO_4$, 16 µM NaEDTA, 3.6 µM $FeCl_3$, 2.2 µM $MnCl_2$, 0.9 µM $ZnCl_2$, 0.7 µM $CuSO_4$, 0.65 µM $NaMoO_4$, 0.1 µM $CoCl_2$, $3.69 \times 10^{-10}$ M cyanocobalamin, and $2.96 \times 10^{-7}$ M thiamine. Flasks of 10 to 50 mL inoculum cultures were grown photoautotrophically in a $CO_2$ enriched atmosphere (0.2%) under constant 60 µmol m$^{-2}$ s$^{-1}$ photosynthetically active radiation (PAR) using fluorescent tube lighting on a shaker table oscillating at 150 rpm. Experimental algal cultures were grown in Simulated Algal Growth Environment (SAGE) photobioreactors equipped to control light, temperature, and pH conditions. Two levels of illumination were employed throughout experimentation with the low light (LL) condition applied by a single, side-oriented, LED light panel (Reliance Laboratories, Port Townsend, USA) set to deliver 60 µmol m$^{-2}$ s$^{-1}$ PAR. High light (HL) conditions were set to 1000 µmol m$^{-2}$ s$^{-1}$ PAR by adding a second LED panel on the opposite side of the reactor and adjusting the power output of the first panel. Light penetrance into the cylindrical reactors was verified by sampling 21 positions within increasingly dense *P. celeri* cultures using a Walz US-SQS/L spherical micro quantum sensor (Walz, Effeltrich, Germany). All experiments were conducted under dilute conditions to ensure high light exposure unless otherwise noted. With addition of the second light panel, light penetrance was attenuated by <5% in cultures with a chlorophyll density of 0.5 µg mL$^{-1}$ and by ~10% with 1.0 µg mL$^{-1}$ chlorophyll. Therefore, cultures were either inoculated at appropriately low densities or diluted daily by media replacement to maintain a maximum of 0.75 µg mL$^{-1}$ chlorophyll concentration.

### RNA isolation and sequence analysis

Culture sample volumes were adjusted based on optical density measurements to ensure a minimum of 5 µg of chlorophyll was collected into a large volume syringe directly from SAGE photobioreactors. Algal cells were rapidly concentrated onto 0.45 µm pore size syringe filters (PES, Celltreat, USA) and flow through was discarded. The cell sample was rinsed immediately by flowing an additional 5 mL of Tris buffered saline (Bio-Rad, Hercules, California, USA) to remove excess salts for downstream processing. Gene transcription

was halted by quenching in liquid nitrogen and storing filters at −80 °C prior to subsequent extraction steps. The time between removing a sample from the lighted reactor and quenching was limited to <30 s, as previous studies have found certain plant transcript levels to change significantly within 1 min of light intensity changes[29]. Frozen cells were transferred from filters by backflushing ice-cold BE70 buffer, consisting of 70% ethanol with 1% glycerol 0.5% glacial acetic acid in a 0.5X PBS buffer[68], using a syringe similar to a previously described method[69], and deposited into tubes containing RNase-free stainless-steel beads (BioSpec, Bartlesville, OK, USA) on dry ice. Cells were pelleted by centrifugation for 3.5 min at 10k × $g$ in 1 °C, supernatant was discarded, and cells were immediately frozen in liquid nitrogen. The frozen biomass was dry milled using a tissuelyzer (Qiagen, Germantown, Maryland, USA) with the blocks pre-cooled to −80 °C to reduce heat-induced RNA degradation, and then lysed samples were returned to liquid nitrogen. Samples were suspended in extraction reagent from the MasterPure Yeast RNA Purification Kit (LGC, Biosearch Technologies, Hoddesdon, UK), with 300 µL buffer for each sample amended with 2.0 µL proteinase K (50 µgµL$^{-1}$) and 6.75 µL dithiothreitol (DTT; 2 M) for RNase inhibition. The remaining steps were performed according to manufacturer protocol including DNase I treatment for 15 min at 37 °C.

Read data were aligned to the set of annotated transcripts created for the TG2 strain by Synthetic Genomics (Supplementary file; pico_024561_nuc_p0_v3.0.0.cds.fa) using Salmon software (v1.4.0)[70] in selective alignment mode using the "ValidateMappings" argument, and with the full genome (NCBI accession number: JAACMV000000000 https://www.ncbi.nlm.nih.gov/nuccore/1811192891)[71] used as a decoy sequence file to limit spurious mapping of reads derived from unannotated genomic regions. All gene loci or related transcript sequences throughout this study were referred to by unique annotation numbers with the "EMRE3EUKG" or "EMRE3EUKT" prefixes, respectively. Original transcript sequences and associated annotation data related to the genome assembly accession above were generated by Synthetic Genomics using their "Archetype" sequence analysis pipeline as previously described[72] and are also provided as a Source Data file. The transcript count data were subsequently processed by DEseq2[73] using default parameters for estimations of size factors and dispersions for samples and transcripts respectively. Comparisons of transcript level changes between LL and HL conditions were estimated by calculating the Wald statistic on negative binomial distribution models. Subsequent tests for transcripts demonstrating responses to both time in HL and strain genetic background were conducted by a likelihood ratio test (LRT) of variance resulting from isolating the interaction term of the full statistical model; ~ strain + sampling time + strain: sampling time.

Gene clusters were determined by longitudinal analysis of transcript level changes over time with respect to the strain genetic background during HL acclimation using the degPatterns function in DEGreport package (v. 1.26.0)[74]. Significantly similar genes were clustered based on a Kendall test statistic calculated from a correlation distance matrix of all gene transcript levels between groups of replicate data set across the time course. Over representation of gene function categories were computed by comparing Kyoto Encyclopedia of Genes and Genomes (KEGG) ontology (KO) assignments among a given gene set to the KO assignments across the entire genome by a hypergeometric distribution method[75], as implemented in the enrichKEGG function of the R clusterProfiler package[76]. Transcription factor binding site (TFBS) analyses were conducted using the CiiiDER java software suite[77]. In the absence of empirical transcription factor binding site data specific to *P. celeri*, the entire JASPAR database[78] CORE plant position frequency matrices set (accessed July, 2023) was used to scan the 1500 bp upstream region of a given set of genes. Scans of all putative regulatory regions were executed using a deficit threshold value of 0.15, and subsequent enrichment analyses implemented Fisher's exact test to quantify the significance of over or under

representation of each TFBS in terms of proportion of gene regulatory region bound relative to a background gene set constructed from the remaining *P. celeri* genome.

## Genetic engineering

**Preparation of MYB1 transformation construct.** Transformation of *P. celeri* follows published procedures leveraging nourseothricin selection[4]. A colony of *P. celeri* TG2 was boiled in Triton TE (5 mM EDTA, 1 mM Tris, 1% Triton-X100) at 95 °C for 20 min to extract genomic DNA for use as template (1 ul) to amplify the native TG2 MYB1 locus. The 3688 base pair promoter, MYB1 CDS (EMRE3EUKT4064225), and terminator were amplified using NEB Q5 Hot Start High-Fidelity 2X Master Mix Polymerase (New England Biolabs, M0494) and primer set MM01 - MM02 (Supplementary Table 9). The PCR product was purified using Invitrogen PCR Purification Kit (Invitrogen, K310001) and cloned into pGAPDHnat plasmid vector cut with SpeI and SmaI using the Takara In-Fusion Cloning 5x Enzyme kit (Takara Bio, USA) following manufacturer protocol. The pGAPDHnat vector is a derivative of pAK11D[4] and confers resistance to nourseothricin (cNAT). The assembled plasmid was transformed into NEB 10-Beta competent *E. coli* cells (New England Biolabs, C3019) following manufacturer protocol and grown on Luria Broth plates with 100 μg mL⁻¹ ampicillin. Construct assembly and insertion was confirmed with primer set MM03 and DD235 (Supplementary Table 9) and subsequent sequencing from Genewiz/Azenta (New Jersey, USA). The assembled plasmid was named pGAPDHnat-MYB1 (Supplementary Fig. 3). DNA and PCR products were run on 1% agarose gels for 30 min at 130 V.

## Transformation of *P. celeri* TG1

Transformation protocol was adapted from Krishnan et al., 2020[79], omitting the Cas9-sgRNA ribonucleoprotein steps and electroporating with 5 μg DNA instead of 3 μg. *P. celeri* TG1 cells were grown in MASM media and harvested at an OD$_{750}$ of 0.34. Using the approximation $\frac{cells}{\mu l} = 7.0 \times 10^4 * OD_{750}$, and the equation $\frac{3 \times 10^9 cells}{(7 \times 10^7 cells/mL)(A_{750})} = mL\ culture$, $3 \times 10^9$ cells were spun down at 4000 rpm for 10 min and resuspended in refrigerated 375 mM sorbitol. The resuspension was transferred to eight 2 ml microcentrifuge tubes and spun down at 8000 × g for 3 min at 4 °C. The pellets were consolidated and washed with sorbitol with these conditions until one tube with final resuspension concentration of $5 \times 10^8$ cells/100 μl remained. 100 μL of the final resuspension ($5 \times 10^8$ cells) was added to a pre-cooled 2 mm gap cuvette (Fisherbrand, FB120) containing 5 μg pGAPDHnat-MYB1 cut with AflII and placed on ice for 1.5 min. The cells were electroporated using a Gene Pulser II (Bio-Rad, USA) using the Time Constant Protocol with the following parameters: 25 ms, 1.3 kV, 2 mm cuvette. The electroporator delivered a pulse of 1282 V for 24.4 ms, resulting in a capacitance of 50 μF and 500 Ω resistance. The electroporated suspension was immediately transferred to 1 ml of MASM media and incubated at 25 °C with 1% CO$_2$ and 60–70 μmol m⁻² s⁻¹ light for 6 h. After the recovery period, the electroporated culture was added to 11 mL of F/4 top agar overlay[79] and 100 μg mL⁻¹ nourseothricin. Top agar overlay was maintained at 45 °C before introducing transformed cells, mixing, and evenly dividing among four F/4 EU plates, previously described[79], with 100 μg mL⁻¹ nourseothricin. Plates were incubated in the following conditions: 25 °C with 1% CO$_2$ and 60–70 μE light. Colonies appeared 2–6 weeks after electroporation.

## Confirmation of mutation

Colonies that survived on plates with nourseothricin were tested for insertion of pGAPDHnat-MYB1. Colonies were boiled in 20 μL of Triton TE (5 mM EDTA, 1 mM Tris, 1% Triton-X100) for 20 min at 95 °C and 1 ul was used as template for PCR with Onetaq 2X MM HotStart QuickLoad Polymerase (New England Biolabs, M0488) and primers MM03 and DD235 to amplify a 1427 bp product bridging the region of the MYB1

promoter and the selective marker which would be unique in transformant strains. PCR products were run on 1% agarose gels for 30 min at 130 V. PCR products were sent to Azenta/Genewiz (New Jersey, USA) for sequencing to confirm amplicon identity. Confirmed mutants were cryopreserved by collecting culture samples during growth phase in liquid media, -1.0 OD750, followed by addition of 5% DMSO and cooling in a Mr. Frosty™ freezing container (ThermoFisher Scientific Inc., Waltham, MA, USA) to −80 °C. Frozen cultures were archived by storage in a liquid nitrogen cryogenic storage tank for future revival and use. Out of 264 colonies arising on the transformation recovery plates, 4 were shown to harbor the correct expression construct for an overall transformation efficiency of 1.5%. Transgene insertion localization in the TG1-MYB99 mutant line was determined using whole genome sequence data generated from DNA isolated using the same method for RNA isolation described above on biomass pelleted from liquid culture and application of RNase I treatment in place of DNase I. Raw paired end read data were generated by Azenta (New Jersey, USA) on an Illumina MiSeq instrument resulting in >17,000,000 paired-end 250 bp reads. The read data were mapped to the transformation construct sequence resulting in >1000X coverage on average using the BWA-MEM algorithm (v. 0.7.17-r1188)[80]. Reads that overlapped the AflII cut site where the plasmid was originally linearized were collected using SAMtools and assembled into a contig with SPAdes (v. 3.15.5)[81]. This chimeric sequence was utilized as a query for BLASTn alignment to the TG2 reference genome assembly to identify putative insertion sites.

## Transient ¹³C-bicarbonate labeling experiment

*P. celeri* strains TG1, TG2, and TG1-MYB99 were adapted to HL intensity (1000 μmol m⁻² s⁻¹ PAR) by ensuring chlorophyll concentration was maintained below 0.75 μg mL⁻¹ for several days prior to any transient labeling experiment. A 2 L volume was inoculated on the day prior to the transient labeling experiment in the SAGE photobioreactor and total organic carbon (TOC) measurements were taken 2 h before and at the start of the experiment to determine the specific growth rate. OD$_{750}$ of the cultures were always maintained <0.3 for the transient labeling experiment. The culture was then split equally into 6 photobioreactor positions, each of which served as a time point for the transient labeling experiment. Transient labeling was achieved by turning off the pH control (CO$_2$ supply) and by the addition of NaH¹³CO₃ (Cambridge Isotope Laboratories, Andover, MA) up to a final concentration of 11.76 mM simultaneously with 4-(2-hydroxyethyl)-1-piperazineethanesulfonic acid (HEPES) buffer (pH -5) (Sigma Aldrich, St. Louis) up to a final concentration of 5 mM. NaH¹³CO₃ was preferred to ¹³CO₂ to avoid gas-liquid mass transfer limitations and achieve rapid equilibrium. Preliminary experiments were performed to determine the concentration of NaH¹³CO₃ and HEPES buffer that need to be added simultaneously to achieve a step change from ¹²C to ¹³C, maintain pH <7.5, as well as be able to minimize contribution of unlabeled carbon in the labeling dynamics.

Once the ¹³C pulse was introduced, each identical culture was harvested at different time points, namely 0 (no label pulse), 30, 60, 180, 300, and 600 s by rapidly filtering on 9 cm Fisherbrand glass microfiber filters followed by washing with 15 mL of 0.2 M ammonium bicarbonate in 5% methanol kept in an ice bath and quenching in liquid nitrogen. These time points were chosen to capture the complete labeling dynamics of the rapidly labeling CBB cycle metabolites. It was consistently observed that quenching was achieved not more than 10 s after the desired time point. Filters were stored in −80 °C freezer prior to extraction. Experiments were performed in biological triplicates to determine standard deviation in labeling dynamics.

## Metabolite extraction and analysis

Filters with biomass were kept on dry ice and 26 μL of 100 μM 1-¹³C Leucine (Cambridge Isotope Laboratories, Andover, MA) was added as

the internal standard. The label 1-$^{13}$C Leucine was chosen as the internal standard since contribution of 1-$^{13}$C Leucine from the transient labeling experiment was minimal in this time scale of the experiment. The filters were then folded and stored in 15 mL centrifuge tubes on dry ice. Extraction was performed by first adding 2 mL of precooled methanol followed by maceration till the filter was a pulp. This was followed by addition of 2 mL pre-cooled chloroform and further maceration. Once the extraction was complete with the methanol:chloroform mixture, 5 mL chloroform was added followed by 1 mL of 0.05% ammonium hydroxide (pH ~ 10.4) for phase separation. The extracts were then gently vortexed for a minimum of 15 s followed by centrifugation at 4 °C at 2500 g. The clear phase on the top was then removed and filtered using a syringe filter, diluted in acetonitrile (ACN) (3 ACN: 1 Extract) and transferred to LC-MS vials prior to injection.

The metabolites were separated using hydrophilic interaction chromatography (HILIC) using a BEH Amide column via a gradient method developed prior to injection into an ultra-high resolution Thermo Scientific Q-Exactive mass spectrometer. Retention times of metabolites is reported in Supplementary Table 10. Supplementary Information, Supporting Methods describes further details.

### Instationary Metabolic Flux Analysis (INST-MFA)

Metabolic fluxes were estimated using INCA (v 2.2) software package implemented in Matlab (R2022a)[82]. INCA utilizes transient elementary metabolite unit (EMU) balances to simulate isotopic label enrichment and relies on least squares regression of measured isotope label measurements and extracellular fluxes (for example, biomass sink). INST-MFA involves minimization of sum of squared residuals (SSR) between simulated and measured data to yield intracellular fluxes.

The metabolic reaction network for metabolic flux analysis was constructed to include CBB, pentose phosphate pathway, photorespiration pathway, TCA. The model includes compartmentalization such as chloroplast, cytosol, and mitochondria, and transporters were modeled to allow metabolite transfer between compartments. Dilution parameters are also modeled to allow for instances of inactive metabolic pools or phenomena such as metabolite channeling that can lead to unlabeled metabolite fractions[17]. Biomass production was modeled using reactions for lipid, chlorophyll, protein, carbohydrate, and nucleotide synthesis. Each strain was modeled using a distinct protein biosynthesis reaction that relies on the amino acid composition of protein (Supplementary Table 6 and Supplementary Fig 16.) that was quantified using a validated UHPLC-MS method for algae (Supplementary Fig 17). INCA provides a statistical goodness of fit by a Chi square statistical test as well as confidence intervals for all parameters using a parameter continuation approach. Best fit estimates for each strain were computed with at least 100 repetitions with different random initial guesses to obtain a fit. Each flux map was based on data measured from three biological replicates. Supplementary Information includes all the assumptions for INST-MFA, assumptions utilized for developing biomass production reactions, the entire reaction network including atom transitions (Supplementary Data 3), experimentally measured transient isotopologue enrichment data (Supplementary Data 2), flux estimations, and confidence intervals (Supplementary Data 4), for all the strains studied to recreate the model.

### Chlorophyll fluorescence measurement and analysis

Photophysiology characteristics were determined by fast repetition rate fluorometry (FRRf) using a light-induced fluorescence transient (LIFT) instrument (Version LIFT-SUB-1.0, Soliense Inc., New York, USA) to measure chlorophyll fluorescence (ChlF). Samples were collected from actively growing bioreactor cultures, immediately diluted to <0.4 µg mL$^{-1}$ of Chl in growth media as needed to ensure response within the dynamic range of the instrument, and 3 mL was transferred to the measurement cuvette. ChlF measurements were collected after

5 min of dark incubation followed by a series of measurement flashes under a programed actinic light levels. Measurement flashes were collected with a rapid single turnover (ST) protocol wherein an excitation phase was induced by 175 blue light flashlets using a 445 nm LED calibrated to deliver 30,000 µmol m$^{-2}$ s$^{-1}$ for a duration of 1.6 µs with a 2.5 µs delay followed by a relaxation phase consisting of 127 additional flashlets with a 20 µs delay time increasing at an exponential rate of 1.025 as defined by the following equation; $j_i = 10^{1.6 + 1.025 \times i} \mu s$. Where $j_i$ is the delay interval length of the $i^{th}$ flashlet, and resulting ChlF was detected at 685 (±10) nm for each flashlet to construct a transient curve for each complete flash. The kinetics of photosystem II were estimated from each flash data set by fitting a model with three free parameters over 200 iterations as described previously[83]. The blue ST flash was followed by a green ST flash with the same flashlet timing using a 535 nm LED delivering 46,000 µmol m$^{-2}$ s$^{-1}$ in order to estimate functional absorption cross sections with minimized packaging effects caused by blue light excitation. Samples were subsequently incubated for 5 additional min in darkness before applying a multiple turnover (MT) measurement flash consisting of 6000 flashlets of the green LED for a duration of 1.6 µs with a 20 µs delay determining maximal fluorescence of fully saturated PSII ($F_m$). The same MT flash protocol was delivered under increasing actinic light up to 2050 µmol m$^{-2}$ s$^{-1}$ (Supplementary Fig 14.) and the resulting $F_m'$ values were used to calculate NPQ by the equation; $(Fm−Fm')/Fm'$. Photosynthetic electron transport chain inhibitor experiments were conducted under the same FRRf protocol by diluting 50X stock solutions into 3 mL of cell suspension samples to final concentrations of 33 µM and 10 µM of Octyl Gallate and Antimycin A, respectively immediately prior to the dark incubation phase.

### Reporting summary

Further information on research design is available in the Nature Portfolio Reporting Summary linked to this article.

## Data availability

The transcriptomic data generated in this study have been deposited in the NCBI SRA database under accession code PRJNA1024918. The genome assembly data used in this study are available in the NCBI database under accession code JAACMV000000000 [https://www.ncbi.nlm.nih.gov/nuccore/1811192891]. The metabolomics data generated in this study have been deposited in the Metabolomics Workbench database under accession code ST003035 [https://doi.org/10.21228/M8P728]. Supplementary Data 4 was used to make Fig. 3. All remaining source data used to make figures are included in a separate source-data file, the source data for Supplementary Figs. and analysis code is available upon request, with no conditions required to access the data and scripts aside from contacting the authors. Source data are provided with this paper.

## Code availability

Scripts used for RNAseq analysis specifically are available at GitHub [https://github.com/ssteiche/pico_rnaseq].

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

## Acknowledgements

We are grateful for the technical assistance of the NREL Bioenergy Algae Analytical team, Stefanie Van Wychen, Alicia Sowell and Bonnie Panczak for compositional characterization of algal biomass and total organic carbon analysis; Steven Rowland for HPLC-MS method development; Peter Shanta for initial method development on metabolite extraction; The Viridos Inc. bioinformatics team; Drs. Scott Becker, Laura de Boer, and Eric Moellering, is gratefully acknowledged for the initial assembly and annotation of the *Picochlorum* TG2 genome that was used throughout this work. Protocol optimization for the FRRf measurements was aided by helpful interactions with Dr. Zbigniew Kolber. Technical troubleshooting on transformation experiments from Drs. Anagha Krishnan, Melissa Cano, and Matt Posewitz, at Colorado School of Mines, is gratefully acknowledged. We are grateful for helpful discussions before and during this project with Dr. Kelsey McNeely at ExxonMobil Technology and Engineering Co. (EMTEC). This work was financially supported by a collaborative research and development agreement (CRD-18-00765) with EMTEC. The National Renewable Energy

Laboratory (NREL) is operated for the US DOE under Contract No. DE-AC36-08GO28308. The views expressed in this article do not necessarily represent the views of the DOE or the U.S. Government. The U.S. Government retains and the publisher, by accepting the article for publication, acknowledges that the U.S. Government retains a non-exclusive, paid-up, irrevocable, world-wide license to publish or reproduce the published form of this work, or allow others to do so, for U.S. Government purposes.

## Author contributions

L.M.L., J.W., D.D., L.R.C., S.S., and R.N. conceived the idea and supervised this study; S.S., A.D., M.M., J.L., and D.D. performed the experiments. S.S. and S.B. participated in the genetic data analyses while A.D., S.S., and L.M.L. conducted flux and combined omics analysis, S.S., L.R.C., R.N., J.W. and D.D. contributed to photophysiology method development and analysis. D.D., J.L., E.P.K., and M.M. carried out the genetic engineering, including construct design and protocol optimization. S.S. and A.D. wrote the original draft of the manuscript. All authors reviewed and edited the final draft. L.M.L., R.N., and L.R.C. acquired funding and established the joint project.

## Competing interests

The authors declare no competing interests.
