## [Peer Review File · Nature Communications]

Reviewers' comments:

Reviewer #1 (Remarks to the Author):

This study applied Transcriptional analysis and ¹³C-MFA to reveal microalgae metabolism under High Light Stress. This research is extensive and may have high significance. However, the ¹³C-MFA part has major weaknesses.

Similar work has been done extensively in cyanobacterial species with similar insights. However, ¹³C-MFA of eukaryote algae is very challenging due to micro-compartments. Many metabolites are shared between cytosol and mitochondrion or Chloroplast. The labeling of these metabolites could not be used to constrain INCA model. In addition, precise INCA analysis requires the determination of metabolite pool sizes, which is also difficult to do. In my opinion, it is impossible to obtain a reliable flux network by using their methods.

Another issue for the ¹³C-MFA work is shading effect. The experiment uses very large volume (2L /6) and such volume may cause the subpopulation that lives in dark area due to biomass shading. Their experimental approach is problematic. During INCA labeling process, Biomass filtration and washing process will stress cells and cause artifacts since metabolite turnover is very fast and can be a few seconds or minutes.

Reviewer #2 (Remarks to the Author):

The most significant result from the authors was the identification of a transcription factor that was related to the higher growth rate of one strain of *P. celer* vs another strain of *P. celer*. The large-scale control of transcriptional regulation was validated by transgenic expression of MYB99 in the TG1 strain.

The biomass composition change between TG1 and TG2 is amazingly high under HL. This indicates the defect in the TG2 strain, as its protein content drops from ca. 50% to only 30%, while its carbohydrate content increases from 10% to 45%.

Suggested revisions.

The introduction was very long and read as if it came from a DOE grant proposal. It should be shortened and more focused on the fundamental processes studied in this work. The promise of how the results apply to crop improvement are too much of a stretch.

Line 347- this conclusion is a stretch. were the experiments on the other organisms performed in exactly the same way as this study?

Table 2: The text mentions DNA and RNA were assumed from previous work in algae and chlorophyll content was assumed. The Table has error bars which indicates experimental measurements. Was it an assumption or measured? If assumed, it should be indicated in the table.

Line 497- the fact that the flux model for TG2 could not be explained by measurement error is a significant cause for concern. I recommend repeating the biomass composition measurements. If the model didn't provide a statistically acceptable fit, how can you present the data in Figure 3?

Line 538-540. The difference in precursor is quantitatively different, but not actually different precursors. I assume TG2 still makes carbohydrates in the chloroplast and TG1 still makes protein in the cytosol. The main point is the fluxes are altered, which is obvious as the model is asked to fit to different biomass compositions.

The result in lines 576-579 is expected. Under light, the TCA cycle doesn't need to produce energy in the form of NADH or ATP as it comes from light.

Line 609- Given the difference in biomass composition, you should calculate the higher NADPH demand in TG2 vs TG1. I see you did this in the next section. Please connect the thoughts between the sections.

Reviewer #3 (Remarks to the Author):

The manuscript by Steichen et al is well written and informative. Comparative analysis between two phenotypically distinct cultivars of *P. celeris* is warranted. The authors identify a potential transcription factor that likely plays an important role in cellular metabolism and photosynthetic activity. Additionally, they provide new insights into *P. celeris* metabolomics. Several issues should be addressed to make the manuscript more informative. Differential expression of CCA and generation of a new transgenic line are central to the manuscript but more development of the CCA functional story is necessary. More thoroughly describe the differences between CCA in TG1 and TG2. Specifically, are there amino acid changes in the coding sequences? If so, are they in potentially critical AA for potential function? Gene alignments are provided in the Suppl but interpretation is important. Alternatively, are the most

significant changes restricted to UTRs? It is unclear whether the primary difference in TG1-MYB99 is gene dosing, expression or functional diversity and this should be explained more clearly. In general, more analysis of CCA would be beneficial to the manuscript. Ideally, the authors should make a knockout of the native CCA in TG1-MYB99 and in the WT backgrounds to more clearly define the role of CCA if feasible. The manuscript would benefit from doing diel transcriptomics on CCA to further probe its potential role in circadian control if feasible. TG1 and TG1-MYB99 have similar CCA transcript levels, but very different LL phenotypes. More detailed hypotheses regarding the observed outcomes would benefit the manuscript.

The authors have provided important new information regarding the growth phenotypes of *P. celer* cultivars (and transformants) and have identified a potentially important transcriptional regulator. Additionally, they provide important new transcriptomic and proteomic data that are comprehensive and will be of significant importance to other researchers. Additional context and potentially providing more clarity on the physiological function of CCA would be beneficial.

PAM and FRRf experiments are not described in the Methods, please add.

Minor items:

Line 47: There is a double period around “.3.” Additionally, describe this transcription factor in more detail in the text.

Line 155: Describe BE70 buffer.

Line 229: what is “op”? top?

Line 235: % of Triton?

Line 265: HEPES pH?

Line 276: fix -80

Line 355: Genebank Accession numbers should be provided for EMRE.... denoted genes (or database guidance provided) here and throughout the manuscript so that the community can access these sequences.

Line 369: are there other CCA homologs in the *Picochlorum* genome or just this one? If others, how do their transcripts change?

Line 372: Is this in the determined/predicted 3' UTR?

Figure 1: The 4064225 labelling could be more clear. Add box outline to specific location?

Line 399: Should this be “TG2 like”?

Line 528: change “reveals” to “suggests”.

Line 570: “linear branch” instead of “linear cycle”.

Line 639: lower case “synthase”

Figure 4 should not have a line connecting the data points.

Suppl line 107: remove “for”

Suppl data: ATP should be included for lipid biosynthesis.

Reviewers' comments with detailed response:

Reviewer #1 (Remarks to the Author):

- “This study applied Transcriptional analysis and 13C-MFA to reveal microalgae metabolism under High Light Stress. This research is extensive and may have high significance. However, the 13C-MFA part has major weaknesses.”
- “Similar work has been done extensively in cyanobacterial species with similar insights. However, 13C-MFA of eukaryote algae is very challenging due to micro-compartments. Many metabolites are shared between cytosol and mitochondrion or Chloroplast. The labeling of these metabolites could not be used to constrain INCA model. In addition, precise INCA analysis requires the determination of metabolite pool sizes, which is also difficult to do. In my opinion, it is impossible to obtain a reliable flux network by using their methods.”
 - **Response:** It is not clear which insights the reviewer refers to here. Throughout the manuscript we consistently refer to existing literature and the agreements in flux between various phototrophic organisms. We build on extensive previous literature on the application of INCA to eukaryotic organisms, where the compartmentalization is well-described and modeled. Although similar work has been done in cyanobacteria, we believe the insights here are novel. Our main findings show that the triose phosphate transporter (TPT) plays a crucial role in carbohydrate and growth phenotypes are likely under the control of the CCA1 transcription factor. Cyanobacterial flux maps do not include compartmentalization like eukaryotic algae and thus transporter conclusions and findings are not relevant. The TPT3 transporter’s role has independent external validation in *C. reinhardtii* in recent work where TPT3 null mutants showed reduced growth, increased carbohydrates and light sensitivity, but included no flux data (Huang et al., *Plant Cell*. 2023 Jun 26;35(7):2592-2614. doi: 10.1093/plcell/koad095. PMID: 36970811; PMCID: PMC10291034.), much like what our novel flux data reveals.
 - We agree that eukaryotic labeling experiments are challenging due to compartmentalization and shared metabolite pools, as experimental data only represents ‘pooled’ labeling dynamics. We address this issue by utilization of an appropriate, well-documented, computational approach as previously done in algae (Treves et al., *Nat. Plants* **8**, 78–91 (2022). <https://doi.org/10.1038/s41477-021-01042-5>). This high impact work also did not include absolute quantitation of pool size in the INCA modeling, rather assigning these as free parameters to be fit by the model. We have updated the flux models and obtained new statistically-acceptable fits for TG2 and TG1-MYB99. The updated data is very similar to previous data, did not result in change in magnitude of fluxes between the three strains, and did not result in different interpretation. The previous lack of fit was not a sign of the incorrect model (since TG1 has a statistically acceptable fit), but more due to very low experimental error on mass isotopomer distribution (MID) data (some of which was as low as 0.001). However, such low error estimates may not account for all error sources. For example, mass isotopomer fractions obtained from orbitrap instruments can be biased so that minor isotopomers are underestimated (*Sundqvist et al., 2022*, *PLOS Computational Biology*, doi.org/10.1371/journal.pcbi.1009999). Therefore, for TG2 and TG1-MYB99, we increased the minimum standard deviation of MID data to 0.01 (1%) and 0.02 (2%) respectively, to obtain statistically acceptable fits. Assumption of minimum standard deviation for MID data is commonly used in previous and recent flux studies:
 - Treves et al., *Nat. Plants* **8**, 78–91 (2022),
 - Yu King Hing et al., 2019 <https://doi.org/10.1016/j.ymben.2019.08.014>,

- Nakajima et al., *Plant and Cell Physiology*, Volume 58, Issue 3, March 2017, Pages 537–545, <https://doi.org/10.1093/pcp/pcw233>)

Often, such parameter relaxation modifications are not disclosed clearly in manuscript methodology. For example, the most well described state of the art flux models in cyanobacteria have used a minimum standard deviation of 0.03 (3%), which is higher than that required by our model to obtain a fit (Abernathy et al., 2017 *Biotechnol Biofuels* **10**, 273 (2017). <https://doi.org/10.1186/s13068-017-0958-y>). Figure 3, and all the associated main and supplementary data has been updated accordingly.

- “Another issue for the 13C-MFA work is shading effect. The experiment uses very large volume (2L /6) and such volume may cause the subpopulation that lives in dark area due to biomass shading. Their experimental approach is problematic. During INCA labeling process, Biomass filtration and washing process will stress cells and cause artifacts since metabolite turnover is very fast and can be a few seconds or minutes.”
 - **Response:** We meticulously controlled for shading effects and cell physiology far beyond other documented methods and described this explicitly several times (most notably in new lines 113-117, 232-234), reporting that light attenuation was kept below 5% in dilute cultures to ensure the relevance of this work. This necessitated larger volumes for the reported metabolomics and transcriptomics work, which our cultivation systems are set up for. With this set up, metabolite samples during dynamic labeling were quenched in less than 10 seconds (line 276), among the fastest reported. Further, biomass filtration has previously been used for harvesting cells for metabolic flux experiments (Jaiswal et al., *Plant J.* 2023 Oct;116(2):558-573. Doi: 10.1111/tbj.16316) and may also prevent metabolite leakage when compared to alternate methods such as quenching with prechilled methanol (Sake et al., *Biotechnol Prog.* 2020 Sep;36(5):e3015. doi: 10.1002/btpr.3015).

Reviewer #2 (Remarks to the Author):

- “The most significant result from the authors was the identification of a transcription factor that was related to the higher growth rate of one strain of *P. celer* vs another strain of *P. celer*. The large-scale control of transcriptional regulation was validated by transgenic expression of MYB99 in the TG1 strain.”
- “The biomass composition change between TG1 and TG2 is amazingly high under HL. This indicates the defect in the TG2 strain, as its protein content drops from ca. 50% to only 30%, while its carbohydrate content increases from 10% to 45%.”
 - **Response:** We confirmed that the compositional shifts are most pronounced in TG1, whereas TG2 does not alter its carbohydrate content upon the same light transition experimental conditions.

Suggested revisions.

- “The introduction was very long and read as if it came from a DOE grant proposal. It should be shortened and more focused on the fundamental processes studied in this work. The promise of how the results apply to crop improvement are too much of a stretch.”
 - **Response:** We have reviewed the introduction and edited for clarity and brevity (New lines: 42-45; 55-84).

- “Line 347 (New lines: 344-346)- this conclusion is a stretch. were the experiments on the other organisms performed in exactly the same way as this study?”
 - **Response:** The reviewer points to experimental distinction between reported results of other studies. We made sure to match the significance cut offs (p-value and L2FC) to define the respective responsive genes. The supplemental table 3 includes some of the physiological conditions that underpinned the reported work (as far as they were available), i.e. light level (μE) and time in high light (min), culture density was not always included.
 - The larger fraction of genes identified in *P. celer* in our work could be related to the nature of our experiments avoiding light attenuation above 0.5 μg per liter.
- “Table 2: The text mentions DNA and RNA were assumed from previous work in algae and chlorophyll content was assumed. The Table has error bars which indicates experimental measurements. Was it an assumption or measured? If assumed, it should be indicated in the table.”
 - **Response:** The DNA and RNA, and chlorophyll content of biomass were calculated to satisfy mass balance of the biomass prior to flux modeling, and the assumed values are based on extensive prior data collected in our laboratory, illustrating that the quantitation of these components in the biomass do not dramatically vary between species and physiological conditions (Van Wychen 2023) (New line 459-464) and subsequently scaled to meet mass balance (described in in great detail in **Supplemental Materials Fluxomics, INST-MFA assumptions, point #6**). We have removed the standard deviation from the table to distinguish from measured values for FAME, protein and carbohydrates. The standard deviation for DNA, RNA and ChlA was originally included as the calculated variance between replicated, scaled values. Updated table is shown below:

Strain	Light	Growth Rate (hr^{-1})	FAME (%)	Protein (%) ^a	Carbohydrate (%)	DNA (%)	RNA (%)	Chl A (%)
TG1	LL	0.110 ± 0.038	18.8 ± 0.2	55.9	10.2 ± 0.2	3.9	8.7	2.6
TG2	LL	0.079 ± 0.039	21.6 ± 1.3	49.0	8.6 ± 0.8	4.8	10.7	3.2
TG1-MYB99	LL	$0.048 \pm 0.037^{\text{b}}$	19.6 ± 0.1	55.0 ± 0.3	10.4 ± 0.4	3.8	8.6	2.6
TG1	HL	0.220 ± 0.079	11.8 ± 0.2	29.9 ± 0.1	44.9 ± 0.3	3.4	7.66	2.3
TG2	HL	0.323 ± 0.043	16.9 ± 0.3	50.8	16 ± 1.1	3.9	8.6	2.6
TG1-MYB99	HL	0.254 ± 0.041	9.5 ± 0.3	21.0 ± 3.0	56.1 ± 2.7	3.4	7.6	2.3

- “Line 497- the fact that the flux model for TG2 could not be explained by measurement error is a significant cause for concern. I recommend repeating the biomass composition measurements. If the model didn't provide a statistically acceptable fit, how can you present the data in Figure 3?”

- **Response:** We have updated the flux models and obtained new statistically-acceptable fits (passing chi-squared test) for TG2 and TG1-MYB99. The updated data is very similar to previous data and did not result in change in magnitude of fluxes between the three strains, and did not result in different interpretation. The previous lack of fit was not a sign of the incorrect model (since TG1 has a statistically acceptable fit), but more due to very low experimental error on mass isotopomer distribution (MID) data (some of which was as low as 0.001). However, such low error estimates may not account for all error sources. For example, mass isotopomer fractions obtained from orbitrap instruments can be biased so that minor isotopomers are underestimated (*Sundqvist et al., 2022, PLOS Computational Biology, doi.org/10.1371/journal.pcbi.1009999*). Therefore, for TG2 and TG1-MYB99, we increased the minimum standard deviation of MID data to 0.01 (1%) and 0.02 (2%) respectively, to obtain statistically acceptable fits. An assumption of minimum standard deviation for MID data is commonly used in previous flux studies (Treves et al., *Nat. Plants* 8, 78–91 (2022), Yu King Hing et al., 2019 <https://doi.org/10.1016/j.ymben.2019.08.014>, Nakajima et al., *Plant and Cell Physiology*, Volume 58, Issue 3, March 2017, Pages 537–545, <https://doi.org/10.1093/pcp/pcw233>) and often not disclosed clearly and kept under the wraps. For example, the most well described state of the art flux models in cyanobacteria have used a minimum standard deviation of 3%, which is higher than that required by our model to obtain a fit (Abernathy et al., 2017 *Biotechnol Biofuels* 10, 273 (2017). <https://doi.org/10.1186/s13068-017-0958-y>). Figure 3, and all the associated main and supplementary data has been updated accordingly.
- “Line 538-540 [New lines: 544-547]. The difference in precursor is quantitatively different, but not actually different precursors. I assume TG2 still makes carbohydrates in the chloroplast and TG1 still makes protein in the cytosol. The main point is the fluxes are altered, which is obvious as the model is asked to fit to different biomass compositions.”
 - **Response:** We have updated the lines to clarify this statement as: “The different biomass composition of the strains necessitates a distinct carbon allocation (**Table S5, Supplemental Materials Fluxomics**). TG2, which has a higher protein content demands more precursors from the TCA cycle whereas TG1 and TG1-MYB99 produce carbohydrates and require higher glycolytic flux in the chloroplast.”
- “The result in lines 576-579 [New lines 589-591] is expected. Under light, the TCA cycle doesn't need to produce energy in the form of NADH or ATP as it comes from light.”
 - **Response:** We have modified the text as follows: “This could potentially be a mechanism in fast growing strains under high light conditions to prevent loss of CO₂ from the conversion of AKG to succinate via the succinyl CoA as energy demands in the form of NADH or ATP are met by light reactions.”
- “Line 609 (New lines 627-629)- Given the difference in biomass composition, you should calculate the higher NADPH demand in TG2 vs TG1. I see you did this in the next section. Please connect the thoughts between the sections.”
 - **Response:** We have modified the text at the start of section 3.6 to better connect the sections as follows: “Due to distinct biomass composition and metabolic demands for

these strains, the cellular ATP and NADPH demands were calculated based on metabolite fluxes from our model (Supplemental Materials Fluxomics, Table S9).”

Reviewer #3 (Remarks to the Author):

- “The manuscript by Steichen et al is well written and informative. Comparative analysis between two phenotypically distinct cultivars of *P. celeris* is warranted. The authors identify a potential transcription factor that likely plays an important role in cellular metabolism and photosynthetic activity. Additionally, they provide new insights into *P. celeris* metabolomics. Several issues should be addressed to make the manuscript more informative. Differential expression of CCA and generation of a new transgenic line are central to the manuscript but more development of the CCA functional story is necessary.”
- “More thoroughly describe the differences between CCA in TG1 and TG2. Specifically, are there amino acid changes in the coding sequences? If so, are they in potentially critical AA for potential function? Gene alignments are provided in the Suppl but interpretation is important. Alternatively, are the most significant changes restricted to UTRs? It is unclear whether the primary difference in TG1-MYB99 is gene dosing, expression or functional diversity and this should be explained more clearly. In general, more analysis of CCA would be beneficial to the manuscript.”
 - **Response:** The density of variant sites between the wild type strains was significantly higher in the UTR regions surrounding the CCA1 coding sequence. This observation along with the dramatically different changes in transcript levels in response to high light led us to hypothesize that the dosage effect was most critical to the observed variations otherwise. We have added a brief description of the amino acid sequence variation between TG1 and TG2, along with a supplemental figure addition (**New lines 371-389; Fig. S3B**). Currently, the only evidence for functional significance of specific residues in the PcCCA1 sequence is alignment to the NCBI conserved domain database, which defined a short, conserved motif related to the SANT myb-like DNA binding family. None of the documented conserved amino acid sites were found to vary between TG1 and TG2 CCA1 proteins, as thus we chose not to focus on this aspect during our experiments. Our lab does have an interest in further defining the CCA1 related regulatory network in *P. celeris*, and we are currently pursuing a collaboration to conduct DAP-seq analyses with the goal of validating predicted binding targets across the genome.
- “Ideally, the authors should make a knockout of the native CCA in TG1-MYB99 and in the WT backgrounds to more clearly define the role of CCA if feasible. The manuscript would benefit from doing diel transcriptomics on CCA to further probe its potential role in circadian control if feasible. TG1 and TG1-MYB99 have similar CCA transcript levels, but very different LL phenotypes. More detailed hypotheses regarding the observed outcomes would benefit the manuscript.”
 - **Response:** We do agree that knock out lines would be of extremely high interest. We have dedicated significant effort (27 repeated independent transformation experiments) to generate knock out lines by CRISPR-targeted engineering, without success. While not

conclusive, a lethal phenotype of a knock out line would be consistent with our not finding a successful transformant. From the literature, we do know that in other organisms, including *Arabidopsis* it is possible to generate knock-out mutants for CCA1 and LHY1 respectively and in combination, suggesting that lethality may not be certain. Alternatively, it is understood that *Arabidopsis* has significantly more redundancy in circadian clock-related genes and functions compared to a more compact genome like *P. celeri* where only one version of this transcriptional regulator was found. Continuing the functional analysis of the CCA1 regulator is part of ongoing and future work in our laboratory

- “The authors have provided important new information regarding the growth phenotypes of *P. celeri* cultivars (and transformants) and have identified a potentially important transcriptional regulator. Additionally, they provide important new transcriptomic and proteomic data that are comprehensive and will be of significant importance to other researchers. Additional context and potentially providing more clarity on the physiological function of CCA would be beneficial.”
 - **Response:** BBX32 gene expression effects on soy bean productivity provide an example of modulation of CCA1 can result in significant alteration in light response leading to changes in biochemical composition on the whole plant level. The *P. celeri* strains here demonstrate that these effects can be linked with carbon fixation rates and directly with dramatic compositional shifts outside of the context of reproductive cycle adjustments.
- “PAM and FRRf experiments are not described in the Methods, please add.”
 - **Response:** This was an oversight, we have updated the manuscript and all details of the chlorophyll fluorescence FRRf experiments have been added to the methods section of the main manuscript document (New lines 291-316). Additional details are referred to in supplemental figure 7 defining the custom protocol development.

Minor items:

- Line 47: There is a double period around “.3.” Additionally, describe this transcription factor in more detail in the text.
 - **Response:** The double period has been removed. Additional details about the transcription factor are added as shown in the highlighted text in the updated manuscript (New lines 371-389; Fig. S3B) as follows; “The coding sequence of the TG1 CCA1 gene contains up to 20 predicted nonsynonymous sequence variants with respect to the TG2 genomic locus. Only two changes in amino acid residues lie within the single identifiable conserved functional domain region, neither of which alter the conserved residues defining the SANT domain structure (**Fig. S3B**).”
- Line 155: Describe BE70 buffer.
 - **Response:** Added description on lines 128-129
- Line 229: what is “op”? top?
 - **Response:** Yes, it is “top”. We have corrected the spelling error.
- Line 235: % of Triton?
 - **Response:** The Triton solution was described in section 2.3.1, and so that has been copied into this line as well (new line 210)

- Line 265: HEPES pH?
 - **Response:** pH was 5.15, respective text is updated (new line 242)
- Line 276: fix -80
 - **Response:** It has been corrected to -80° C.
- Line 355: Genebank Accession numbers should be provided for EMRE.... denoted genes (or database guidance provided) here and throughout the manuscript so that the community can access these sequences.
 - **Response:** We are currently working with our industry collaborators to fully release the genome annotation data. We have obtained permissions to publish GFF files containing the EMRE denoted gene loci related to the assembly housed at NCBI. As we do not have direct access to update the original NCBI accession, we are awaiting a complete upload and release of currently proprietary information. The NCBI record will be updated, and we expect the completion within the next month.
- Line 369: are there other CCA homologs in the Picochlorum genome or just this one? If others, how do their transcripts change?
 - **Response:** There is a single copy of the CCA1 gene in the *P. celeri* genome. As mentioned in the text it does share some similarities with LHY genes in NCBI. We hypothesize that this single gene is performing a dual CCA1/LHY role, similar to the LHL2 gene in Glycine max.
- Line 372: Is this in the determined/predicted 3' UTR?
 - **Response:** This is likely near the border of the true 3' UTR. Our RNA-seq read data map out to ~150-170 bp beyond the stop codon. This caveat has been included in the revised manuscript (New lines 373-374)
- Figure 1: The 4064225 labelling could be more clear. Add box outline to specific location?
 - **Response:** Figure 1 has been updated and 4064225 location has been identified by adding a box outline.
- Line 399: Should this be "TG2 like"?
 - **Response:** This is absolutely correct, and has been updated (New line 405)
- Line 528: change "reveals" to "suggests".
 - **Response:** "reveals" to "suggests" correction made.
- Line 570: "linear branch" instead of "linear cycle".
 - **Response:** "linear branch" instead of "linear cycle" correction made.
- Line 639: lower case "synthase"
 - **Response:** Corrected
- Figure 4 should not have a line connecting the data points.
 - **Response:** We believe that connecting lines are appropriate for these data as they are derived from time series measurements on samples under increasing actinic light. This seems to be typical in other publications such as: Shou et al., 2022 *Plant Physiology* 188, 2 (1264-1276)
- Suppl line 107: remove "for"
 - **Response:** It has been removed as suggested.
- Suppl data: ATP should be included for lipid biosynthesis.
 - **Response:** ATP has been included in lipid biosynthesis reaction. The table S9 has been updated with the reaction: "9 AcCoA + 16 NADPH + 8 ATP-> C18 FATTY ACID + 9 CoA + 8 ADP + 16 NADP+". The data has been updated to reflect this change in Table S9., Table S5., as well as line 620 in the main text. None of the conclusions were altered because of this change.

REVIEWERS' COMMENTS

Reviewer #2 (Remarks to the Author):

The authors have identified an algae with dramatically varying phenotypes. In their revised manuscript the flux maps are now statistically acceptable fits. I appreciate the edits and the authors have satisfactorily answered all major concerns.

Reviewer #3 (Remarks to the Author):

The authors have successfully addressed my concerns from my first review.